# STRategies to manage Emergency ambulance Telephone Callers with sustained High needs: an Evaluation using linked Data (STRETCHED) – a study protocol

Rabeea'h W Aslam [1], Helen Snooks,[1] Alison Porter,[1] Ashrafunnesa Khanom,[1] Robert Cole,[2] Adrian Edwards [3], Bethan Edwards,[1] Bridie Angela Evans [1], Theresa Foster,[4] Rachael Fothergill [5], Penny Gripper,[1] Ann John [1], Robin Petterson,[6] Andy Rosser,[2] Anna Tee,[1] Bernadette Sewell,[7] Heather Hughes,[1] Ceri Phillips [7], Nigel Rees [8], Jason Scott [9], Alan Watkins [1]

**Correspondence to**
Dr Rabeea'h W Aslam;
r.w.aslam@swansea.ac.uk

## ABSTRACT

**Introduction** UK ambulance services have identified a concern with high users of the 999 service and have set up 'frequent callers' services, ranging from within-service management to cross-sectoral multidisciplinary case management approaches. There is little evidence about how to address the needs of this patient group.

**Aim** To evaluate effectiveness, safety and efficiency of case management approaches to the care of people who frequently call the emergency ambulance service, and gain an understanding of barriers and facilitators to implementation.

**Objectives** (1) Develop an understanding of predicted mechanisms of change to underpin evaluation. (2) Describe epidemiology of sustained high users of 999 services. (3) Evaluate case management approaches to the care of people who call the 999 ambulance service frequently in terms of: (i) Further emergency contacts (999, emergency department, emergency admissions to hospital) (ii) Effects on other services (iii) Adverse events (deaths, injuries, serious medical emergencies and police arrests) (iv) Costs of intervention and care (v) Patient experience of care. (4) Identify challenges and opportunities associated with using case management models, including features associated with success, and develop theories about how case management works in this population.

**Methods and analysis** We will conduct a multisite mixed-methods evaluation of case management for people who use ambulance services frequently by using anonymised linked routine data outcomes in a 'natural experiment' cohort design, in four regional ambulance services. We will conduct interviews and focus groups with service users, commissioners and emergency and non-acute care providers. The planned start and end dates of the study are 1 April 2019 and 1 September 2022, respectively

**Ethics and dissemination** The study received approval from the UK Health Research Authority (Confidentiality Advisory Group reference number: 19/CAG/0195; research ethics committee reference number: 19/WA/0216).

## Strengths and limitations of this study

► Use of privacy-protected routinely collected anonymised linked data will provide an efficient, inclusive and holistic picture of service use.
► Impact likely to be as high as the study is closely aligned to current health and social care policy agenda.
► Public Patient Involvement throughout the study.
► Quasi-experimental study design is not as strong as prospective randomised trial.
► Quantitative health outcomes are based on routine data with no self-reported quality of life outcomes.

We will collate feedback from our Lived Experience Advisory Panel, the Frequent Caller National Network and Research Management Group for targeted dissemination activities.

## INTRODUCTION
### Background

Pressures on emergency ambulance services are growing at an unsustainable rate in Europe and North America.[1–3] The escalation of emergency calls to the ambulance services presents as a major operational challenge.[4 5] The volume of emergency calls to ambulance services in England doubled from 4.72 million in 2001/2002 to 9 million in 2014/2015,[1] and national performance targets were not met for 32 months consecutively.[6]

Emergency ambulance services provide care for those with urgent and life-threatening health conditions. Although most people make few calls to the emergency ambulance service, a small minority of patients

make more intensive use.[7] Unresolved health or social care needs within this group put additional pressure on already stretched emergency ambulance services, which were originally designed to respond to patients with time-critical needs for clinical intervention, rather than to manage non-acute care needs.

Definitions have varied[8 9] but UK ambulance services agree that people who make five or more calls per month or 12 calls over a 3-month period should be classified as 'frequent callers'. In London, of a total of 1.7 million calls made during 2014–2015, 1622 people met 'frequent caller' criteria and generated 49 534 ambulance attendances, at an estimated cost of £4.4 million to the ambulance service.[1]

Research shows that people who make high use of emergency services also experience higher mortality rates than those with lower use of services.[10] People who call the emergency ambulance service frequently are often vulnerable,[4 7 11 12] more likely to be from low socioeconomic group live alone, experience mental health challenges, including self-harming behaviour, live with chronic conditions and have increased chances of falling.[13 14] Repeated access of services and calling may be because problems are unresolved or patients are not aware of an alternative pathway to access appropriate care.[7 15]

Taking lessons from initiatives in primary care to incentivise the prioritisation of care of frail older patients who are at risk of emergency admission to hospital,[11] commissioners in the UK now require ambulance services to have management strategies in place for people who call frequently.[10 12] In some areas, case management approaches are in place, with comprehensive care plans developed for patients who call the emergency ambulance service frequently, through multidisciplinary, cross-sector management groups. Key decision-makers from commissioning, acute, primary, secondary and charitable health and social care providers meet with ambulance service staff to share, stratify risk and manage patients in regular multidisciplinary team meetings.[16]

The introduction of case management has the potential to support a change to safe and equitable out-of-hospital care for this patient group, and to avoid patients being shifted to another part of the emergency care system, wider NHS or social care without their care needs being addressed, but research is needed to evaluate this new approach.

## Research aim
The aim of the study is to evaluate effectiveness, efficiency, safety and patient experience of case management approaches to the care of people who call the emergency ambulance service frequently, and gain an understanding of barriers and facilitators to implementation.

## Research objectives
► Develop an understanding of predicted mechanisms of change to underpin the evaluation.

► Describe epidemiology of people who use the 999 service frequently.
► Evaluate case management approaches to the care of people who call the 999 ambulance service frequently in terms of:
  – Further emergency contacts (emergency ambulance calls, emergency department (ED) attendance and emergency admissions to hospital). (Note to copywriter: Formatting here and below needs to be amended - as per that on page 2 under Research Objectives)
  – Effects on other health and social care services.
  – Adverse events (deaths, injuries, serious medical emergencies and police arrests)
  – Costs of intervention and subsequent use of health and social care
  – Patient experience of intervention and subsequent use of health and social care
► Identify challenges and opportunities associated with using case management models, including features associated with success, and develop theories about how case management works in this population.

## METHODS AND ANALYSIS
### Study design
STRategies to manage Emergency ambulance Telephone Callers with sustained High needs—an Evaluation using linked Data (STRETCHED) is a mixed-methods 'natural experiment' evaluation. We will use anonymised linked routine outcomes and qualitative data in four UK ambulance services with one intervention and one control site in each service.

### Work package 1: logic model
We will develop a logic model in consultation with stakeholders during a face-to-face workshop to underpin the evaluation and inform data collection and finalisation of outcome selection. The logic model will include definition of key components of case management in this setting, predicted mechanisms of change and possible outcomes, positive or otherwise. Stakeholders to be invited to this workshop include clinical and managerial partners working within the ambulance service, EDs, primary care and high-intensity user services from across the UK.

Figure 1 shows a draft logic model, developed from the research literature and specialist knowledge of research team members, to be used as a basis for discussion and development during work package 1 (WP1).

### Work package 2: evaluation
We will undertake an evaluation using a natural experiment cohort design in four ambulance services, using quantitative anonymised linked routine data to describe epidemiology and assess effects of the intervention on processes, outcomes, safety and costs of intervention and subsequent health and social care up to 6 months later, with adjustment for covariates, including prior service

**16/47116/64 STRETCHED - STRategies to manage Emergency ambulance Telephone Callers with sustained High needs – an Evaluation using linked Data**
**Emergent LOGIC Model**

**Situation:**
High dependency on 999 services by vulnerable patients who call 999 repeatedly is characterised as inappropriate use of resources and ineffective patient care. Research in this area is limited. It is not known what works and for whom?

**Goal:**
To ensure that this patient group receives appropriate and effective care which reduces their dependency on 999 ambulance services for care

**Population:**
People who call ambulance services repeatedly with complex unmet care needs

**Setting:** Ambulance services in Wales and England who use varying models of care. This can range from full case management to a 'list approach' -flagging up people who call often. Case management involves collaboration with wider health economies (primary, secondary and community care and social care) to manage the care of this patient group

**Inputs:**

Organisational support

Resources – staff, time

Training for staff

Availability of appropriate services to refer patients to

Ambulance service policy and protocols

Information sharing Protocols

Casemix, typology of repeat callers to the 999 service

**Predicted Mechanisms of Change:**

Assessment of crisis intervention

Information sharing

Case management forums

Case managers -including management of case literature and care plan

Recruitment of patients Time to discharge

Contact with patient- face to face, telephone, home visits, Letters

Location of contact – GP surgery, hospital, community based location

Type of support- Therapy
Welfare (housing, income)
Access to information Transition between care services

What works for whom? When?

The patient group is diverse, will responses be flexible and varied?

**Outputs (Process Measures):**

Reduction in the number of patients who call frequently and/or reduction in the number of calls per person

Reduce patient mortality

Reduce costs

**Patient reported outcomes:**

Improve patient access to appropriate care

Improve patient experience of care

Improve patient involvement in designing care plans

**Mediating factors and resources**
* Ambulance services willing to adopt new models of care
* Information sharing across wider health economies
* Acceptability of alternative care pathways, other than ED, by patients and carers
* Organisational culture within ambulance services and wider health economies
* Funding for new models of care

Synthesis of evidence and data analysis

**Figure 1**   Draft logic model. ED, emergency department; GP, General Practitioner.

use. We will also collect qualitative data from focus groups and interviews in each intervention site about the views and experiences of stakeholders (commissioners, emergency and non-acute health and social care providers) regarding acceptability, successes and challenges of case management approaches for this group of patients, and in-depth interviews with a range of service users within the target population.

### Work package 3: synthesis of quantitative and qualitative findings

We will formally synthesise quantitative and qualitative findings from work package 2 (WP2), informed by the logic model developed in WP1 and in consultation with stakeholders included in the research team.

### Setting

The study will be undertaken in four ambulance services, identified through a survey of practice across the UK, where both case management and traditional 'within-service' models are in place in different areas within the service. Emergency ambulance services in the UK which provide emergency are to '999' callers which is free at the point of use. Following mergers in the 1990s, services cover large areas, each serving a population of between 3 million and 9 million people; most cover urban and rural locations and provide a range of emergency responses, including telephone advice, attendance of an emergency vehicle for face-to-face assessment by a paramedic or emergency medical technician, and conveyance to hospital for further care when judged clinically appropriate.

Characteristics of control and intervention site care in the four ambulance service sites included in STRETCHED are summarised in table 1.

### Outcomes
► Further emergency contacts:
– Emergency ambulance calls.
– ED attendance.
– Emergency admissions to hospital.

– Declassification/reclassification as 'frequent caller'.
► Effects on other health and social care services.
► Adverse events, as available:
– Deaths.
– Injuries.
– Serious medical emergencies.
– Police arrests.
► Costs of intervention and subsequent use of health and social care.
► Patient experience of intervention.

### Data collection

We will use a parallel cohort 'natural experiment' study design to determine effects on processes and outcomes of care, using anonymised linked data from NHS Digital and the NHS Wales Informatics Service (NWIS). The variation in exposure and outcomes will allow us to carry out analysis to link effects to intervention, that is, causes.[1] We will include callers meeting the criteria for classification as 'frequent caller' by the ambulance service during 2018. We expect to recruit 300 high users per service (n=1200), allowing us to detect a change in rate of further emergency events/death of ±20% with 90% power and 95% confidence (see figure 2). We will finalise data items following completion of the logic model but expect to include (up to 6 months): further calls to emergency ambulance service, ED attendances, emergency admissions and deaths; declassification or reclassification as 'frequent caller'; costs; and details of demographics, case mix and patterns of calls, for example, 'out of hours' (evenings/nights/weekends/holidays). Historical data about prior service use will allow us to adjust analyses for differences between cohorts, strengthening this study design.

| Table 1 | Sites within participating services and key features of control (usual) and intervention (case management) site care | |
| --- | --- | --- |
| | **Control site: usual care** | **Intervention site: new model of care** |
| Care models: generic description | Aim: to reduce or stop calls made by people making high use of the 999 service<br>Within-service management<br>► Letter sent to GP and patient<br>► Callers flagged; care management plan developed for use in ambulance call centre to triage patient when call comes in<br>► Contact may be made with other services to intervene<br>► Police action may be requested | Aim: to address and meet complex patient needs, thereby reducing emergency contacts by people making high use of the 999 service<br>MDT cross-sector case management<br>Usual (within-service) care plus<br>► Dedicated frequent caller nurse/prehospital care practitioner funded by CCG/HB<br>► MDT meetings attended by ambulance service; partnership approach with other agencies, including district nursing, social workers, police, out of hours providers, mental health professionals, ED, voluntary sector, NHS 111, GPs and occupational therapy to develop and share joint care plans to address patient need |

CCG, Clinical Commissioning Group; ED, emergency department; GP, General Practitioner; HB, Health Board; MDT, multidisciplinary team.

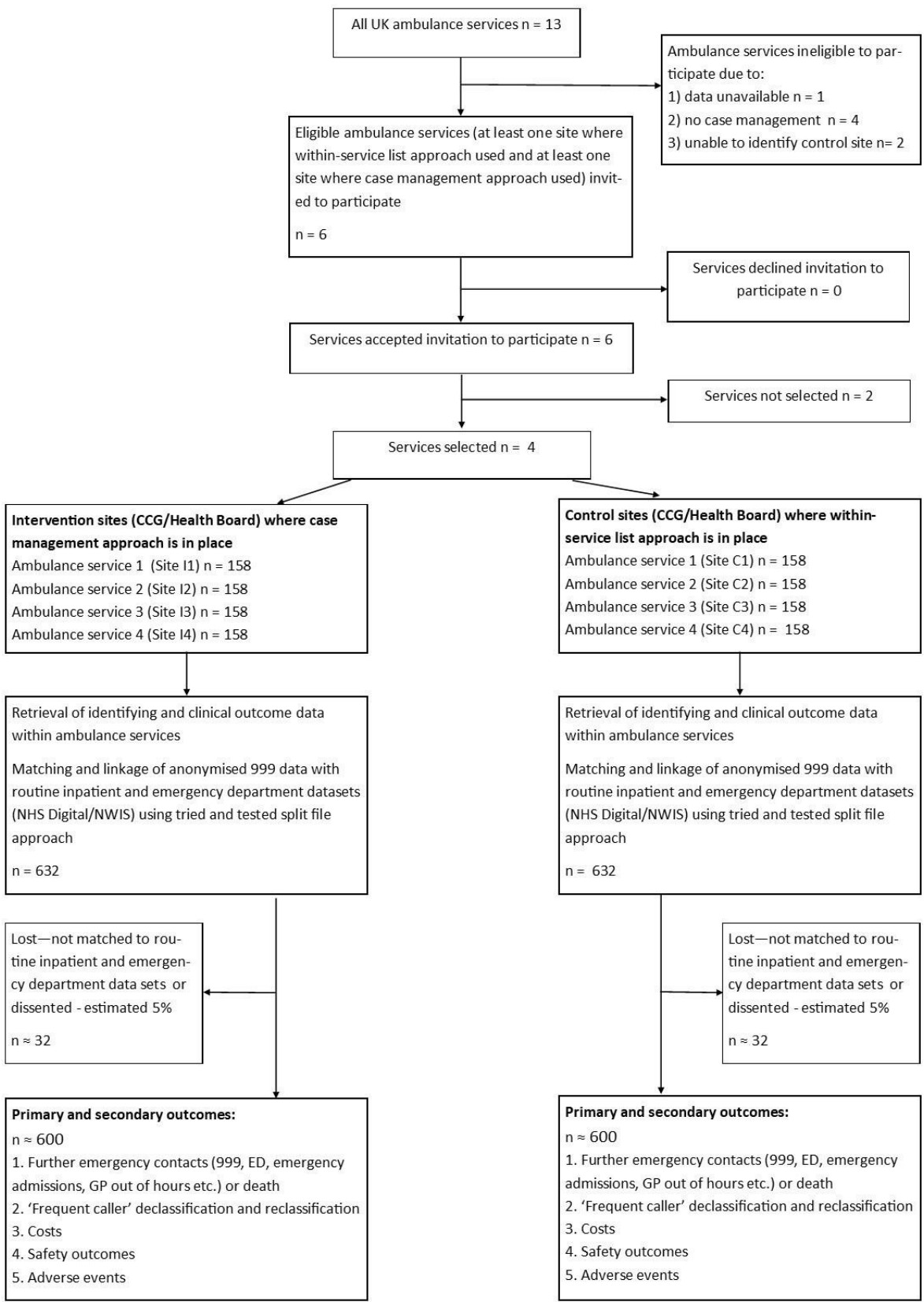

**Figure 2** 'Natural experiment' flow diagram. ED, emergency department; GP, general practitioner; NWIS, NHS Wales Informatics Service.

We will collect qualitative data at the intervention site within each participating ambulance service to explore perceptions of how the intervention works, what creates its effect (if any), why it might function differently in different settings and for different groups of people, and any challenges to implementation and delivery of the intervention. Subject to any restrictions associated with COVID-19, we will conduct four focus groups (supplemented where necessary by telephone or online interviews) with stakeholders—including frontline staff (paramedics and call handlers), partner health and social care providers, commissioners and managers (n=36). Focus group topic guides will cover case management delivery processes, barriers and facilitators to changed working, perceived impact for patients, issues around diversity and terminology, strengths and weaknesses of the approach and wider organisational impact across health economies such as information sharing, communication, and continuity of care.

To ensure we gain a strong picture of patient experience and circumstances, we will conduct largely unstructured interviews with people in intervention areas of each service who are referred for case management (n=32). Research paramedics based at each site will undertake interviews as far as possible. We will use this method to provide an opportunity for patients to provide their own narrative about their circumstances, experiences and views regarding their needs, service use and care received before and after the case management intervention, and the intervention itself, including terminology used. We will work with participating services to identify and invite callers to this key patient-focused element of the study. We will sample purposively to include a wide range of views and experiences, and select individuals with differing demographic characteristics, length of time they have required care and management approach provided for them to include typical and atypical patient stories. We will try to include patients from black, Asian and minority ethnic groups.

With respondents' consent, we will audio-record and transcribe all individual and group interviews.

## Data analysis
### Quantitative analysis
Ambulance services will provide NHS Digital and NWIS with information on recruited patients, to allow matching to anonymised data from multiple sources. We will create a single integrated study database for analysis within the Secure Anonymised Information Linkage (SAIL) Gateway.[17]

We will summarise patient recruitment via a Consolidated Standards of Reporting Trials flowchart,[18] and will provide descriptive data summaries for patients, by service and by site, for demographics, call patterns and volume. Numbers of events will be dichotomised where contextually useful (for instance, to declassify or reclassify callers), and converted to rates when appropriate.

Our primary analyses will be by treatment allocated, so that patients meeting the criteria for 'frequent caller' status in an intervention (case management) site will be included in analysis in that arm of the study whether they were offered or received any intervention. The outcome measure time point will be 6 months from a patient's first appearance on a monthly update to the 'frequent callers' list. This will apply to patients at control sites as well as intervention sites. This time point is appropriate to allow us to detect any effect, as case management is a targeted, time limited intervention designed to work within 6 months.

We will use multilevel generalised mixed linear models to obtain adjusted comparison of outcomes in patients at intervention sites versus those in patients receiving usual care at control sites. Using linked routine data will allow us to gather retrospective data on service use for callers included in each cohort, strengthening comparisons by enabling adjustment for historical and contemporaneous differences in service use, case mix and demographics. Reflecting an initial focus on comparing (any) case management approach with the service list approach, coding for sites will initially assume that outcomes under usual care are similar across ambulance services, as are outcomes at intervention sites. However, we will use background information on the service interventions, the descriptive summaries and formal testing, based on flexible and nested site coding, to test and modify this assumption, as appropriate—for instance, allowing intervention outcomes to vary systematically from service to service. The precise form of models will reflect the nature of the variable under consideration (logistic models for binary variables, negative binomial models for count variables and linear models for raw and transformed measurement outcomes, including rates); the multilevel element of the model will reflect the geographical clustering of sites within services. The adjusted comparisons will incorporate information on covariates and factors based on demographic and case-mix data. We will formalise these analyses in a Statistical Analysis Plan, using the relevant Swansea Trials Unit Standard Operating Procedure (SOP), which will detail conventions on model fitting (including inclusion and exclusion rules for covariates and factors), management of missing data and the reporting of outcomes.

### Health economic evaluation
Using a cost–benefit framework, we will compare the implementation cost of case management approaches to the potential benefits of 'avoiding expenditure' in subsequent health and social care resource utilisation from a public sector perspective. The overall resource implications for case management approaches for frequent callers will include intervention implementation costs, costs of 999 calls and the costs arising from utilisation of other health and care services in the 6-month follow-up period.

The costs of case management interventions will be derived through discussion with relevant staff and observation of operational processes and procedures, considering staff time and other resources required to deliver the intervention. Each resource element will be costed using appropriate unit costs derived from published sources, and/or consultation with relevant finance staff.

The costs associated with a sample of the 999 calls in each study area will be determined by identifying staff inputs, along with materials, equipment, therapies and other relevant resources used during the response to the call and the processes involved in its management and completion. A cost profile for 999 calls will be developed for each geographical area and summarised as cost per 999 call.

The utilisation of other health and care services by the study cohorts in the control and intervention sites will be captured by routine data sources (ie, NHS Digital and the SAIL databank), and costed using appropriate published unit costs. The inclusion of mental health and social care resources will be subject to routine data availability and quality validation.

To assess the rate of return on investment, we will describe changes in service utilisation over time, using changes in volume of calls and their occurrence. Our cost–benefit analysis will include net present value and internal rate of return estimates to assess the relative value for money of case management approaches in managing frequent callers of ambulance 999 services and whether the cost of case management services is shared across organisations. We will undertake a series of sensitivity analyses to estimate the effect of parameter variation on baseline findings and to determine the extent to which case management of these callers is an efficient use of public funds.

### Qualitative analysis

Analysis of focus group and interview transcripts will be carried out by members of the research team with the two public and patient involvement representatives, alongside input from the research paramedics. We will remove all identifiable data from interview transcripts before analysis. We will use a data-driven thematic approach to analysis which generates themes from the implicit and explicit ideas within participants' accounts.[19] We will follow the six stages of analysis described by Braun and Clarke[20]: data familiarisation, generating initial coding, searching for themes, reviewing themes, defining, naming themes and producing a report. During the analysis process, we will check emerging coding and themes with the wider research team and members of the patient panel. We will assess data saturation[21–23] during analysis to see whether new themes are emerging by reviewing the codebook (the working document that records updates in changes to codes). We will also assess whether we have an adequate range of participants reflected in our purposive sample. Analysis of the large volume of data (transcripts, field and observation notes) generated by the interviews will be supported by use of NVivo, computer-assisted qualitative data analysis software.

### Synthesis

Synthesis and reporting of quantitative and qualitative findings will be informed by the STRETCHED logic model. Quantitative data will be used to draw conclusions about comparative costs and effects; qualitative data will help us to understand and interpret these results, and to generate theories about how the new models of care are working. We will bring together key themes from across all the work packages on the effectiveness, attitudes, barriers and facilitators to case management. We will interpret overall effectiveness and cost-effectiveness results in the light of analysis from the qualitative data about which components of case management, for example, care plan, timing of interventions and shared decision-making, are perceived to work well and for whom. We will use the logic model to inform the synthesis of results, which will be considered and interpreted at a joint meeting of the Research Management and Patient Advisory Groups.

## ETHICS

We have received approval from the Health Research Authority (19/WA/0216) and NHS R&D permissions at all participating organisations.

Patients will not be approached for consent to participate in the main effectiveness study (WP2) as identifying information will be held within ambulance services only and not shared with the research team. With information governance permissions in place, retrospective routine data will be linked anonymously by NHS Digital and NWIS using a split file approach, for analysis within the SAIL Gateway.[24]

A small sample of current patients who are calling frequently will be invited by the participating ambulance services to give consent and participate in one-to-one interviews. We recognise that these patients may be vulnerable and will take care to ensure that interviews are carried out sensitively, in a place of the respondent's choice or by telephone/online, for example, Zoom and by appropriately trained researchers at each site.

## PATIENT AND PUBLIC INVOLVEMENT

We have worked extensively with public contributors to develop a comprehensive approach to active involvement of patients and the public at all study stages.[23] We are proposing a layered approach, so that people can be involved at strategic and local levels in line with their interest, experience and health. Our aim is to enable active and meaningful involvement throughout to enhance research quality, rigour and ethical standards, in line with the Swansea Trials Unit SOP on service user involvement[23] and best practice.[25] Our public co-applicants (PG and BE) have been actively involved in study design and shaping the proposal. They have been involved in proposal

design to undertake interviews with patients, challenging biases and assumptions in the team and helping choose of proper language to describe people who phone 999. They are members of the Research Management Group and equal partners in decisions about study implementation and dissemination.

We will also convene a Patient Advisory Panel of six individuals recruited through community groups, support agencies and third sector networks. The Panel will reflect the range of people who make frequent calls to 999, including older people, those with chronic illness and people from lower socioeconomic levels. We will seek to ensure that public contributors within the study match as closely as possible the diverse population of people making frequent calls. This Panel will be a less formal route for people to contribute to specific tasks such as advising on patient-facing materials, sense-checking patient results and devising dissemination materials. We will hold face-to-face or virtual meetings and visit people in their homes if that is their preference, to ensure their involvement.[23 25 26]

We will recruit a further two public members to the independent Research Advisory Group to bring patient perspectives to oversight and scrutiny decisions. We will offer honoraria for all involvement and reimburse incurred expenses.

We will provide training for all public contributors, proposing an induction at start of involvement and training in specific skills such as good clinical practice, analysis, meeting skills and dissemination skills, as contributors require.[27 28]

## DISCUSSION

Evaluation of interventions that have already been implemented is challenging. It is impossible to carry out randomised controlled trials in this circumstance, frequently encountered in the real world of health services research.[29] In this circumstance, alternative evaluation designs need to be used, inevitably sacrificing the ability to definitively ascribe changes observed to the intervention being evaluated—as there is always the possibility of confounding by other systematic differences between sites or patients. In the STRETCHED study, we have planned a quasi-experimental study in which we will compare differences between control sites (usual care) combined and intervention (case management) sites combined. Using linked anonymised outcomes, we will be able to adjust for previous patterns of service utilisation, thus strengthening our comparisons.

It is important to provide evidence about what works and how for this patient group, to inform the development of care. We aim, through this study, to gain robust evidence about costs, effects, safety and stakeholder views, including patients, so that the best care can be provided to improve processes and outcomes for people who make high use of the emergency ambulance service, as well as to reduce pressure on services that provide emergency care.

**Author affiliations**
[1]Swansea University Medical School, Swansea University, Swansea, Wales, UK
[2]West Midlands Ambulance Service NHS Foundation Trust, Brierley Hill, UK
[3]Division of Population Medicine, Cardiff University, Cardiff, Wales, UK
[4]East of England Ambulance Service NHS Trust, Melbourn, UK
[5]London Ambulance Service NHS Trust, London, UK
[6]Welsh Ambulance Services NHS Trust, Saint Asaph, UK
[7]College of Human and Health Sciences, Swansea University, Swansea, UK
[8]Pre-Hospital Emergency Research Unit, Welsh Ambulance Service NHS Trust, Swansea, UK
[9]Faculty of Health and Life Sciences, Northumbria University, Newcastle upon Tyne, UK

**Contributors** The study concept and design was conceived by HS, AW, RWA, AP, AJ, AK, AE, BE, PG, BAE, TF, RF, AR, RP, RC, AT, BS, HH, CP and JS. The work will be conducted and reported by TF, RF, AR, RP, RC, HS, AP, AW, RWA, AJ, AK, AE, BE, PG, BAE, AT, BS, HH, CP and JS. All authors provided edits and critiqued the manuscript for intellectual content.

**Funding** This project is funded by the National Institute for Health Research (NIHR) Health Services & Delivery Research (HS&DR) Programme (reference 18/03/02). The views expressed are those of the author(s) and not necessarily those of the NIHR or the Department of Health and Social Care.

**Competing interests** None declared.

**Patient and public involvement** Patients and/or the public were involved in the design, or conduct, or reporting, or dissemination plans of this research. Refer to the Methods section for further details.

**Patient consent for publication** Not applicable.

**Provenance and peer review** Not commissioned; externally peer reviewed.

**ORCID iDs**
Rabeea'h W Aslam http://orcid.org/0000-0002-0916-9641
Adrian Edwards http://orcid.org/0000-0002-6228-4446
Bridie Angela Evans http://orcid.org/0000-0003-0293-0888
Rachael Fothergill http://orcid.org/0000-0003-1341-6200
Ann John http://orcid.org/0000-0002-5657-6995
Ceri Phillips http://orcid.org/0000-0003-1076-9289
Nigel Rees http://orcid.org/0000-0001-8799-5335
Jason Scott http://orcid.org/0000-0001-7031-2171
Alan Watkins http://orcid.org/0000-0003-3804-1943

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
