## [Reviewer comments · BMJ Open]

ARTICLE DETAILS

TITLE (PROVISIONAL)	STRategies to manage Emergency ambulance Telephone Callers with sustained High needs – an Evaluation using linked Data (STRETCHED): study protocol
AUTHORS	Aslam, Rabeeah; Snooks, Helen; Porter, Alison; Khanom, Ashrafunnesa; Cole, Robert; Edwards, Adrian; Edwards, Bethan; Evans, Bridie; Foster, Theresa; Fothergill, Rachael; Gripper, Penny; John, Ann; Petterson, Robin; Rosser, Andy; Tee, Anna; Sewell, Bernadette; Hughes, Heather; Phillips, Ceri; Rees, Nigel; Scott, Jason; Watkins, Alan

VERSION 1 – REVIEW

REVIEWER	Alanazy , Ahmed University of New England, School of Rural Medicine
REVIEW RETURNED	30-Aug-2021

GENERAL COMMENTS	You have done a great job. I am really interested to see your final work .
--

REVIEWER	McManamny, Tegwyn Monash University, Paramedicine, rural health
REVIEW RETURNED	26-Sep-2021

GENERAL COMMENTS	It was a pleasure to review this protocol, which addresses a very important issue within ambulance and the health system more broadly. I only have minor comments for your consideration:  * The abstract for this piece of work is lengthy, and a more concise version should be prepared if possible. * How will this study ensure that frequent callers who DO require emergency ambulance attendance are provided with that option? * How many patients does the study intend to interview? * There are a range of formatting errors in the reference list that should be corrected.
---

REVIEWER	Stassen, Willem University of Cape Town, Division of Emergency Medicine
REVIEW RETURNED	13-Oct-2021

GENERAL COMMENTS	Thank you for the opportunity to review this study protocol. This work addresses a very important problem faced, not only in high-income countries but also in low- to middle-income countries that have developed emergency medical services. The proposed study incorporates elements of system performance, cost-effectiveness, and patient safety outcomes in order to evaluate case management as an intervention. The study is strengthened by also including control sites, something not
--

	always possible or feasible. The study is further strengthened by the inclusion of the patient's voice through qualitative work and regular patient and public involvement activities. There are just a few minor typographical errors that will need to be attended to.
--	--

VERSION 1 – AUTHOR RESPONSE

Reviewer: 1 Mr. Ahmed Alanazy , University of New England	
You have done a great job. I am really interested to see your final work .	Thank you for your kind words.
Reviewer: 2 Dr. Tegwyn McManamny, Monash University	
It was a pleasure to review this protocol, which addresses a very important issue within ambulance and the health system more broadly. I only have minor comments for your consideration:	Thank you for kind words. We have amended the protocol to add your suggestions.
1. The abstract for this piece of work is lengthy, and a more concise version should be prepared if possible.	Thank you, this has been amended now.
2. How will this study ensure that frequent callers who DO require emergency ambulance attendance are provided with that option?	Implications for policy and practice are beyond the scope of STRETCHED, and we have no direct involvement in formulating policy or practice in any service. Our remit is to obtain and assess evidence on whether or not case management is effective, and (potentially) for whom, and to summarise and present that evidence to a range of stakeholders and interested parties, including policy-makers and service managers. We will aim to ensure that our research findings are widely disseminated through appropriate channels
3. How many patients does the study intend to interview?	As mentioned in page 9 of the manuscript, To ensure we gain a strong picture of patient experience and circumstances we will conduct largely unstructured interviews with people in intervention areas of each service who are referred for case management (n=32)'
4. There are a range of formatting errors in the reference list that should be corrected.	Thank you, these have been corrected now.
Reviewer: 3 Dr. Willem Stassen, University of Cape Town	
Thank you for the opportunity to review this study protocol. This work addresses a very important problem faced, not only in high-income countries but also in low-to middle-income countries that have developed emergency medical services. The proposed study incorporates elements of system performance, cost-effectiveness, and patient safety outcomes in order to evaluate case management as an intervention. The study is strengthened by also including control sites, something not always possible or feasible. The study is further strengthened by the inclusion of the	Thank you for your kind words. I have gone through the protocol again and addressed the errors.

patient's voice through qualitative work and regular patient and public involvement activities.

There are just a few minor typographical errors that will need to be attended to.